# Mechanisms, Machinery, and Dynamics of Chromosome Segregation in *Zea mays*

**DOI:** 10.3390/genes15121606

**Published:** 2024-12-16

**Authors:** Marissa E. Duffy, Michael Ngaw, Shayna E. Polsky, Abby E. Marzec, Sean S. Zhang, Owen R. Dzierzgowski, Natalie J. Nannas

**Affiliations:** Department of Biology, Hamilton College, Clinton, NY 13323, USA; mduffy@hamilton.edu (M.E.D.); mngaw@hamilton.edu (M.N.); spolsky@hamilton.edu (S.E.P.); amarzec@hamilton.edu (A.E.M.); sxzhang@hamilton.edu (S.S.Z.); odzierzg@hamilton.edu (O.R.D.)

**Keywords:** *Zea mays*, chromosome, spindle, spindle assembly checkpoint, kinetochore, mitosis, meiosis, meiotic drive

## Abstract

*Zea mays* (maize) is both an agronomically important crop and a powerful genetic model system with an extensive molecular toolkit and genomic resources. With these tools, maize is an optimal system for cytogenetic study, particularly in the investigation of chromosome segregation. Here, we review the advances made in maize chromosome segregation, specifically in the regulation and dynamic assembly of the mitotic and meiotic spindle, the inheritance and mechanisms of the abnormal chromosome variant Ab10, the regulation of chromosome–spindle interactions via the spindle assembly checkpoint, and the function of kinetochore proteins that bridge chromosomes and spindles. In this review, we discuss these processes in a species-specific context including features that are both conserved and unique to *Z. mays*. Additionally, we highlight new protein structure prediction tools and make use of these tools to identify several novel kinetochore and spindle assembly checkpoint proteins in *Z. mays*.

## 1. Introduction

*Zea mays* (maize) is a foundational model system in molecular biology and cytogenetics, offering unparalleled insight into genetic behavior and chromosomal organization. Barbara McClintock, a pioneering figure in genetics, discovered transposable elements through her work on maize kernel pigmentation throughout the 1940s and 1950s [1,2]. By demonstrating the dynamic nature of the maize genome and developing foundational cytogenetic techniques, McClintock laid the foundation for studies in mutagenesis, gene expression, genomic evolution, and chromosome structure [2]. Additionally, McClintock’s contributions paved the way for advanced sequencing, mapping and microscopy techniques now widely employed in maize research [3,4,5].

As a large, easily cultivated organism with distinct mutable loci, visible phenotypic markers, and the ability to outcross and self-cross, maize has become an invaluable resource for genetic mapping, mutant phenotypic analysis, and evolutionary studies [5]. The full maize genome sequence was first released in 2009, assembled through a Sanger-based BAC method using an inbred B73 variety [5,6]. The current *Z. mays* genome assembly (Zm-B73-REFERENCE-NAM-5.0) is composed of 2.3 Gb base pairs with approximately 33,000 protein-coding genes organized into ten chromosomes [7]. Large-scale sequencing efforts have sought to expand the number of maize varieties with assembled genomes including other inbred varieties including Mo17 [8], A188 [9], PH207 [10], W22 [11], European inbred lines F7, EP1, DK105 [12], sweet corn [13] and tropical varieties SK1 [14] and K0326Y [15]. In addition, 25 nested association mapping (NAM) founder lines [16] and 12 founder inbred lines [17] have been sequenced. The repetitive and complex nature of the *Z. mays* genome has created challenges in genome assembly, but a recent telomere-to-telomere (T2T) assembly on Mo17 has been achieved using Oxford Nanopore Technologies and PacBio HiFi reads that covered each of the ten maize chromosomes with a single contig [18]. Repetitive regions such as centromeres (CentC), telomeres, TAG trinucleotide repeats, and nucleolar regions (45S rDNA) have been successfully assembled on all 10 normal maize chromosomes [18], as well as repetitive knob sequences (knob180 and TR-1) on the abnormal 10 chromosome (Ab10) [19].

High-quality genome sequence has accelerated the development of genome editing technologies in maize. Current efforts have primarily focused on crop trait enhancement, including increasing yields, improving the nutritional quality and content, and increasing resistance to both biotic and abiotic stresses [20,21,22]. Delivery of CRISPR components into maize has been challenging due to limitations in transformation technology which primarily rely on *Agrobacterium tumefaciens* or biolistic delivery methods [23,24] and result in low efficiency of successful modification [25,26]. Research using immature zygotic embryos of maize inbred B104 found an average transformation frequency of 4% for the *Agrobacterium*-mediated method and 8% for the biolistic approach [27]. Genome editing in maize is also limited, as no single genetic transformation method is applicable to a wide variety of maize genotypes [28]. Despite these challenges, CRISPR-mediated genome editing in maize has been successfully used to investigate gene function through knock-out, knock-in, and modulation of expression. CRISPR-modification of *ZmbHLH47* and *ZmSnRK2.9,* including both over-expression and mutation, demonstrated these genes promote drought tolerance in maize [29]. Multiplexed, simultaneous editing of three genes allowed for the investigation of maize male fertility and pollen development [30]. Editing of ZmEPSPS has created the first CRISPR-based herbicide resistance maize [31].

In addition to its breadth of genomic resources and tools, *Z. mays* has a range of cytogenetic and microscopy-based techniques that allow exploration of complex biological processes with greater accuracy and insight. Historically, fixed maize samples have been imaged using a variety of staining and probing techniques [3,32]. Cytogenetic tools including immunofluorescence [33] and fluorescence in situ hybridization (FISH) have facilitated imaging of maize chromosomes [34,35]. FISH methods in maize allow the visualization of a range of elements, from individual genes, including transgenes [36], to the painting of whole chromosomes to identify structural variations [37]. Advances in microscopy have allowed for the development of live cell imaging in maize, enabling observation of cell division, chromosome dynamics, and protein localization [38,39,40]. Maize live cell imaging has been greatly enhanced through the creation of a large collection of fluorescent protein marker lines [41]. The collection includes over 100 lines containing GFP-, CFP-, and RFP-tagged proteins marking a range of biological structures including microtubules, chromosomes, major organelles like the ER, and cell walls. The collection also includes trans-activating lines for tissue-specific expression of fluorescent markers [42]. Live-cell imaging has also been used to monitor the dynamic rearrangement of microtubules, nuclei, the endoplasmic reticulum, and endomembrane compartments during cell division and cell plate formation in maize leaf epidermal cells [43]. By understanding the rearrangement of cellular structures during cell division mechanisms, researchers can better understand maize development, potentially leading to improved crop yields.

The extensive molecular tools, ranging from live-cell imaging to genomic editing, and extensive genomic resources establish *Z. mays* as an excellent model system for modern cytogenetic study. With its relatively large chromosomes, maize allows for spindle–chromosome interactions to be easily visualized by microscopy. Genomic resources support the identification of novel proteins and pathways related to chromosome segregation including spindle assembly, kinetochore composition, and the spindle assembly checkpoint. In this review, we explore the advancements made in our understanding of how *Z. mays* chromosomes are segregated by the spindle and kinetochore machinery, as well as an abnormal chromosome (Ab10) found in maize that breaks Mendelian genetics. The five major topics covered in this review (spindles, Ab10, spindle assembly checkpoint, kinetochores, and investigative tools) are listed in Table 1 along with the major highlights in each topic area. Investigation of maize chromosome segregation is critical for both uncovering mechanisms of inheritance and also creating new genetic technologies for this agriculturally important crop.

**Table 1 genes-15-01606-t001:** Topic areas reviewed and major highlights within each topic area.

Topic	Highlights
Spindles	Different morphology in mitotic vs. meiotic spindles Relatively similar spindle dynamics mitosis vs. meiosis Identified spindle mutants: VKS1, CRK2, EB1, RANGAP1, AM1 See Figure 1 for diagram and Figure 2 for live spindle images
Abnormal chromosome 10 (Ab10)	Three distinct Ab10 types based on TR-1 and knob180 knobs Ab10 types show differing meiotic drive rates Prevalence of Ab10 determined in some but not all landraces See Figure 3 for chromosome diagrams
Spindle Assembly Checkpoint	Five identified proteins: MAD2, Bub1, Bub3, SGO1, Aurora B kinase See Figure 1 for diagram and Table 2 for protein list
Kinetochore	Five identified proteins: CENH3, CENPC, NDC80, MIS12 and KNL1 Estimated 85+ unidentified kinetochore proteins See Figure 1 for diagram and Table 2 for protein list
Molecular Tools and Genomic Resources	Cytogenetic tools: FISH and ImmunoFISH Genomic tools: fully assembled genome (v5, hosted by MaizeGDB) Genetic manipulation: CRISPR and transformation Alignment and homology search tools: AlphaFold (3D structural alignment), BLAST (sequence alignment), Hidden Markov models (HMM) account for positional variation for better alignment

## 2. Maize Chromosome Segregation

With its rich history of cytogenetic research and modern tools for probing live processes, *Z. mays* serves as an excellent genetic model system for studying chromosome segregation. Using these tools, studies have investigated the morphology, composition, and dynamics of the *Z. mays* spindle, the microtubule-based machine that segregates chromosomes, as well as the protein-based kinetochore that assembles on centromeres and connects chromosomes to microtubules. Spindles are crucial for ensuring proper chromosome segregation during mitosis and meiosis (Figure 1). In all eukaryotes, spindles are constructed of microtubules, polymers composed of α- and β-tubulin heterodimers [44]. Microtubules undergo cycles of polymerization at their + ends and depolymerization at their − or + ends, allowing for spindle organization into bipolar morphologies, attachment of chromosomes and segregation during cell division [45,46].

### 2.1. Maize Spindle Structure

The goal of the spindle is to accurately separate chromosomes into each of the two daughter cells (Figure 1B). As a result, all eukaryotic spindles must create a bipolar structure in which microtubules emanate outward from two poles with their + ends attaching to kinetochores and aligning chromosomes on the spindle midzone [45]. Once all chromosomes are properly attached to the spindle in metaphase, depolymerization of the microtubules results in retraction of the chromosomes towards the opposite poles (Figure 1B) [45]. In animal cells, spindles are assembled using a “search and capture” mechanism in which microtubules are nucleated outward from poles defined by a centrosome, a microtubule-organizing organelle [47]. In contrast, plants lack centrosomes and instead assemble their spindles via a “self-assembly” mechanism without a clear organizing center. *Z. mays* utilizes the self-assembly mechanism to organize its mitotic and meiotic spindles, as described in more detail below. Despite maize mitosis and meiosis both utilizing the self-assembly mechanism, the morphologies of these two spindle types are quite different [48,49]. The mitotic maize spindles have a boxy, barrel-shaped structure with broad spindle poles (Figure 2A) [50].

Meiotic spindles, in contrast, are more narrow in width than mitotic spindles with sharply focused poles. Meiotic spindles are approximately 35 µm long [51,52] and 8 µm wide at the metaphase plate (Figure 2B,C) [52]. Mitotic spindles are shorter in length, approximately 15 µm long, but have a similar width as meiotic spindles [53]. Mutations in the *tan1* gene produce even shorter mitotic spindles [53], though their effects on meiotic spindles remain unknown. The sharply focused poles of maize meiotic spindles are attributed to meiotically active kinesin-14A motor protein encoded by *ZmKin6* [52]. Without this kinesin, maize produces a phenotype known as *divergent spindle-1 (dv1)* [54]. Using immunostaining and live imaging, two different *dv1* mutations (nonsense mutation and deleterious mutation) were shown to produce spindle phenotypes, specifically in the failures of focusing and organizing the poles [52,55]. Meiotic spindles also display significant curvature compared to mitotic spindles, sometimes occurring as sickle-shaped compared to the box-like mitotic spindle (Figure 2) [48].

### 2.2. Maize Spindle Assembly Dynamics

Like other plants, *Z. mays* assembles spindles without centrosomes, relying instead on self-assembly mechanisms that utilize pre-existing microtubule arrays [56] and nucleate new microtubules near chromatin [57] Microtubules are organized into a bipolar structure through kinesins and other microtubule-associated proteins [58], often passing through a stage of multipolarity that gradually coalesces into a bipolar structure [59]. The development of transgenic maize lines containing fluorescently labeled tubulin [41] and live-imaging techniques have facilitated the investigation of maize spindle dynamics (Figure 2) [38,40]. Spindle assembly time is similar in both processes, averaging 40 min for mitosis and 45 min for meiosis from breakdown the nuclear envelope to establishment of a bipolar metaphase spindle (Figure 2D) [53,60]. This meiotic spindle assembly time is significantly faster than mammalian oocytes in which the process requires several hours [61,62]. Compared to the mitosis duration of *P. patens* (~10 min), *Arabidopsis* mitosis (40 min), and *Arabidopsis* meiosis (~60 min), maize meiotic spindle assembly is similar, occurring within a comparable time scale [60,63,64,65]. Likewise, spindle disassembly rates are similar in maize mitosis and meiosis, averaging 10–15 min (Figure 2D) [53,60]. After chromosomes are retracted to opposite poles, spindles in both mitotic and meiotic spindles disassemble and form a phragmoplast to support the establishment of the new cell wall (Figure 2D) [38,60]. While disassembly timing is similar, meiotic spindle disassembly shows a distinctive two-step process in which the spindle gradually shortens by depolymerizing, then abruptly collapsing which initiates the expansion of the phragmoplast [60]. This two-state disassembly process has not been observed in maize mitosis.

The acentrosomal spindle assembly in maize is regulated by several critical pathways and structures. In mitotic spindle assembly, knockdown and mutational assays are critical for identifying several crucial regulators. One such mutant found is the varied-kernel-size phenotype (*vks1*) [66]. VKS1 expresses mutant ZmKIN11, a kinesin part of the kinesin-14 subfamily [66]. A single nucleotide polymorphism in the eighth exon resulted in a C to T transition in the *vks1* mutant, leading to a nonsense mutation [66]. The absence of VKS1 caused reduced nucleation of radial microtubules, increased tangled and shapeless spindles, splayed poles, and errors in chromosome segregation, including incorrect attachments, pulled in a skewed and uneven manner, and aneuploidy [66]. Similar analysis of the *crumpled kernel mutant* (*crk2*) mutant revealed reductions in endosperm cell number and size and lethality [67]. CRK2 was found to encode TFCB, a tubulin folding cofactor involved in α-tubulin folding, tubulin dimer formation, and regulation of microtubule dynamics. CRK2 regulates microtubule dynamics through its interaction with three important partners, CCT5, TFCE, and EB1, which were identified through yeast two-hybrid assays [67]. CCT5 is a homolog of the human CCT ε subunit that is involved in α-tubulin folding through its interactions with TFCB [67]. TFCE additionally interacts with this complex, forming the TFCE/α/TFCD/β complex [67]. EB1 is a known microtubule-associated protein that modulates the activity of microtubule polymerization at its + ends [68]. Using live cell fluorescent markers, maize EB1 was shown to localize to growing + ends of microtubules within the mitotic spindle [43,67]. Further, TFCB was found to separate EB1 from microtubule ends, leading to microtubule depolymerization [67]. Additionally, when maize mitotic cells move into prophase, EB1 velocity increases, causing an increase in microtubule growth and shrinkage rates [43]. Similar live studies using fluorescently labeled RANGAP1 showed that this protein is required for proper spindle assembly [43]. RANGAP1, which increases RANGTPase activity, generally initiates proper spindle assembly in mitosis by promoting the release of spindle assembly transcription factors [69].

The positioning of the spindle within the maize cell differs dramatically in mitosis and meiosis. Generally, in plant mitosis, a preprophase band (PPB) is identified that consists of microtubules arrays that form in late interphase under the cell cortex [70]. The PPB has been shown to mark the cortical division site, the location where the cell divides [70,71]. Mitotic spindles assemble perpendicular to the PPB, thus utilizing a spindle positioning mechanism based on the cellular location of the PPB. Regulation of the PPB and spindle positioning involves multiple proteins including MLKS and TANGLED1 (TAN1). Mutations in MLkS (*mlks2*) show a misplaced PPB with time-lapse imaging, specifically defective in premitotic nuclear migration toward the polarized site and unstable positioning at the division site post-PPB formation [71]. Through live imaging, mutant *tan1* displays PPB displacement as a result of aberrant cell shape, suggesting that TAN1 does not directly impact PPB [72]. Rather, *tan1* mutants result in delayed bipolar organization, though cells recovered after ~20 min [73]. Further, live imaging of TAN1-YFP and microtubules at the division site show TAN1 guiding the phragmoplast microtubules for cytokinesis [73].

Mechanisms that position the meiotic spindle are less understood as maize meiotic cells lack a preprophase band [72]. Investigation of meiotic spindle position showed that while spindle assembly tends to initiate within the central volume of the cell, spindle position can occasionally deviate from the center of the cell [51]. The meiotic spindle can respond to offsets by dynamically altering the segregation distance of chromosomes, retracting them in an asymmetric pattern to recenter the chromosomes in equal cell volume [51]. While wild-type maize lacks PPBs in meiosis, mutations in the *am1* gene locus have been shown to produce a PPB in meiosis, specifically meiotic mutants *am1-1, am1-2, am1-485, am1-489* [74]. Further characterization of the *am1* gene revealed that nearly all meiotic processes including recombination, pairing, synapsis, and meiosis-specific cytoskeleton organization require functional *am1* [75]. Without the presence of functional am1, pre-meiotic maize cells initiate a mitotic division including the creation of a PPB; however, the *am1* spindles still retain meiosis-specific morphology with focused poles and longer spindle length [75].

*Z. mays* serves as an excellent model system for meiotic-specific processes such as telomere bouquet formation, chromosomal pairing, synapsis, and recombination. An extensive meiotic mutant collection [48,76] includes mutants such as *desynaptic1* (*dsy1*), *desynaptic2* (*dsy2*), and *absence of first division* (*afd*), which exhibit defective synapses that lead to abnormal chromosome pairing [48]. While this review focuses primarily on the segregation of maize chromosomes, readers seeking detailed information on maize meiotic chromosome pairing and recombination are encouraged to consult [77].

## 3. Abnormal Chromosome 10 and Meiotic Drive

In meiosis, chromosomes are segregated such that homologous chromosomes are separated in meiosis I and sister chromatids separate in meiosis II (Figure 1B). These reductional divisions ultimately create haploid gametes in which there is an equal probability of inheriting any combination of chromosomes. This equal probability of inheritance is the basis of Mendelian genetics and our ability to calculate inheritance probabilities. Meiotic drive is a phenomenon in which inheritance of certain alleles is skewed, with offspring inheriting these alleles at higher than predicted rates [78,79]. Meiotic drive is a rare occurrence, but *Z. mays* has one of the most well-characterized systems of meiotic drive. A recent review by R. Kelly Dawe delves deeply into the history and mechanism of maize meiotic drive [80]; in this review, we summarize the major features of this meiotic drive system.

Maize meiotic drive occurs due to the presence of abnormal chromosome 10 (Ab10) (Figure 3) [81]. Ab10 is a mutated variant of normal chromosome 10 (N10); this variant was first identified in 1942 by Marcus Rhoades due to its easily distinguishable cytological structure. Unlike N10, Ab10 contains large heterochromatic DNA regions called “knobs” that consist of two major repetitive sequences, knob180 and TR-1 (Figure 3A) [82,83]. More specifically, the long arm of Ab10 contains several small TR-1 repeat knobs, followed by a large knob composed of knob180 repeats, and a far distal tip comprised of euchromatin (Figure 3A). This distal euchromatic region plays a crucial role in meiotic drive, as deletion mapping has shown that it contains key meiotic driving functions [79]. In maize plants containing Ab10, the heterochromatic knobs function as neocentromeres, structures that segregate chromosomes more efficiently and selectively than normal centromeres (Figure 3B). Ab10 neocentromeres move 38% faster towards poles than normal centromeres [84]. Additionally, the neocentromeres are functionally active earlier in meiosis than normal centromeres, which allows the Ab10 chromosome to attach to the spindle in a preferential manner (Figure 3B). In female meiosis, only one of the four resulting cells becomes an egg capable of passing on genetic information, as the other three apoptose as polar bodies [85]. The neocentromere on Ab10 is capable of attaching and orienting on the meiotic spindle to preferentially segregate it into the developing egg cell (Figure 3B). This skewed segregation leads to Ab10 transmission rates as high as 83% [86,87]. However, Ab10 is only present in maize varieties, or landraces, at 18–22% [88], and within these populations the average frequency of carrying Ab10 is ~30% [89]. This low overall population frequency is likely the result of associated fitness costs, as plants that are homozygous for Ab10 show reduced pollen viability and lower seed production [88]. Heterozygosity of Ab10 does not appear to have fitness defects, which allows the variant to remain in the population [88].

### 3.1. Ab10 Molecular Mechanism

Heterochromatin knob DNA consists of two major repetitive sequences, knob180 and TR-1. Both knob180 and TR-1 are highly abundant tandem repeat sequences; knob180 is approximately 180 base pairs [87,90], and TR-1 is approximately 350 bp [91]. These repetitive knob sequences are located throughout the maize genome [92,93]. Most maize chromosomes possess knobs of varying sizes [87], and thus the presence of a knob alone is insufficient to produce neocentromere activity [86,94]. In addition to four knobs, Ab10 possesses two gene families not otherwise present in the maize genome that act as neocentromere-activating factors [86,94]. The *Kinesin Drive (Kindr)* [86] and *TR-1 Kinesin (Trkin)* [94] genes encode motor proteins that localize on knob sequences and create neocentromeres that promote segregation to the developing egg cell (Figure 3A,B). The ability of these Ab10 neocentromeres to bind microtubules and move on the spindle is independent of kinetochores proteins, the typical protein bridge structure that connects chromosomes to microtubules (Figure 3B). Ab10 neocentromeres and their activating factors do not contain or recruit kinetochore proteins for segregation [79].

The *KINDR* complex is a cluster of eight tandem copies of the *Kindr* gene which encode a kinesin-14 motor protein, KINDR [86]. The cluster is found at the tip of the chromosome below knob180 and is located in a ~1Mb region [86]. The closest maize homolog of KINDR is *ZmKin11* found on the short arm of chromosome 7 [86]. KINDR has a motor domain that facilitates minus-end-directed movement along microtubules and contains a coiled-coil domain (128 amino acids), typical for kinesins, which likely aids in its dimerization, stabilizing its function as a motor protein [94]. Sequence analysis of the KINDR complex reveals it evolved around 12 million years ago and is currently expanding and homogenizing, likely driven by selection for multiple functional copies within the tandem array [86]. Immunostaining for KINDR protein paired with FISH probes for knob180 (ImmunoFISH) reveals that KINDR co-localizes with knob180 sequences and through this interaction powers the movement of knobs as a neocentromere [86]. Microtubule gliding assays demonstrate that KINDR is an active kinesin capable of movement on microtubules, with an average velocity of approximately 154 nm/s, faster than other maize kinesins such as KIN11 with an average velocity of 79 nm/s [86]. Immunostaining also reveals the Ab10 neocentromere activity does not rely on the presence of kinetochores [95], the proteins that typically connect chromosomes to microtubules. Truncations of the Ab10 chromosome [82,88] and epimutant analysis reveal that deletion or suppression of *Kindr* gene expression is sufficient to render Ab10 unable to perform meiotic drive [86]. RNAi suppression assays shows a correlation between decreasing expression levels of *Kindr* and reduction in meiotic drive [86].

The Ab10 chromosome possesses another kinesin gene, *TR-1 Kinesin (Trkin)* located proximally to the TR1-based knobs [94]. Unlike *Kindr, Trkin* is a single copy gene; sequence analysis shows that it is not a homolog of *Kindr* and exhibits significant sequence divergence [94]. While the gene spans 130kb, the actual coding sequence is only ~1700bp. *Trkin* consists of 19 exons, 14 of which are organized in alternating patterns of short and long introns [94]. The TRKIN protein contains three distinct motifs: a short coiled-coil domain (~54 amino acids), a kinesin-14 motor domain, and a nuclear localization sequence [94]. Microtubule gliding assays confirm TRKIN’s function as a minus-end directed kinesin. TRKIN is considerably slower than KINDR, with an average velocity of only 38 nm/s. ImmunoFISH staining reveals exclusive localization of TRKIN in TR-1 repeat sequences and not knob180 sequences [94]. Neocentromeres composed of TR-1 repeats differ from knob180-based neocentromeres as they appear to mobilize earlier in the cell cycle and tend to cover half the spindle length [94]. However, the contribution and significance of the TRKIN motor to meiotic drive is unclear as some Ab10 variants (e.g., K10 [96]) lack functional copies of *Trkin* yet still exhibit high levels of meiotic drive [94].

### 3.2. Ab10 Types

Within the cytogenetic classification of abnormal chromosome 10, there are three distinct variants: Ab10-I, Ab10-II, and Ab10-III (Figure 3C). All three Ab10 variants differ from N10 due to structural features in the long arm, specifically the region distal to the *Colored-1* (*R1*) gene [81]. Within this genomic region, N10 genes are ordered *White2* (*W2*), *Opaque7* (*O7*), *Luteus13* (*L13*), and *Striated2* (*Sr2*), while Ab10 variants have an inverted gene order of *L13-O7-W2-Sr2* [78]. Due to this inversion, Ab10 does not recombine with N10 beyond the *R1* locus [97], but different variants of Ab10 are capable of recombination with one another [89,97,98]. The three Ab10 variants all contain the TR-1, knob180, and the meiotic driver genes *Kindr* and *Trkin,* but the number and location of the repetitive sequences and the relative positioning of the meiotic driver genes differ in each variant (Figure 3C) [88]. The Ab10-I variant has three TR-1 knobs, followed by the shared region of genes *L13-O7-W2-Sr2*, a large knob180-based knob, and the distal tip. The Ab10-II variant has only one TR-1 knob, followed by a similarly sized knob180 region. In addition to this knob, Ab10-II has a smaller, secondary knob180 region at the far distal tip of the chromosome. Ab10-III is structurally similar to Ab10-I with three TR-1 knobs, followed by a large knob180 region. However, this variant has an additional TR-1 sequence after the knob180 region of the chromosome followed by the euchromatic distal tip [99]. Both Ab10-I and Ab10-II exhibit neocentromere activity, which contributes to meiotic drive. It has been found that Ab10-I and Ab10-II increase meiotic drive with a transmission rate of 70–79% [81,89]. However, recent studies indicate that the range for meiotic drive is broader than expected and ranges from 61.6 to 79.5% in Ab10-I and Ab10-II [88]. Ab10-III also exhibits meiotic drive, but the rate varies greatly depending on the geographic origin of maize variety, with rates ranging from 60.0 to 83% [80,88]. The additional TR-1 region in Ab10-III is thought to enhance its segregation efficiency [100].

In addition to the three Ab10 variants, a separate chromosome 10 variant, Knob Chromosome 10 L2 (K10L2), has been identified [100]. Unlike Ab10 or N10, K10L2 contains only heterochromatic TR-1 knobs [97] and lacks the typical large knob180 region and *Kindr* gene family found on all Ab10 variants [86,94]. Despite the lack of these components, K10L2 exhibits a very low level of meiotic drive, at rates of 51–52% when paired with N10 [94,100]. The 1–2% increase in K10L2 transmission compared to N10 is statistically significant and is demonstrated through three independent large-scale crosses involving scoring of over 10,000 offspring [100]. This low-level meiotic drive indicates that the TRKIN/TR-1 complex may be able to act independently as a neocentromere. However, other research indicates that chromosomes lacking the KINDR/knob180 complex but containing TRKIN/TR-1 cannot induce meiotic drive, suggesting that TR-1 alone cannot support meiotic drive [86]. The K10L2 variant has also revealed that TRKIN/TR-1 complex may act in opposite to the KINDR/knob180 complex because in the cross of K10L2 with Ab10, the meiotic drive capability of Ab10 is reduced [100,101].

## 4. Spindle Assembly Checkpoint in Maize

The spindle assembly checkpoint (SAC) regulates the proper attachment of spindles to kinetochores. SAC monitors the occupation of microtubules on kinetochore proteins and further senses tension that should be created on correctly bioriented chromosomes. Without satisfying the checkpoint, the cell cannot proceed to anaphase. While the proteins and pathways of SAC have been substantially characterized in yeasts and metazoans [96] and the conserved SAC homologs have been identified in *Arabidopsis* [99], far less is known about the SAC in plants, including *Z. mays* [102]. Several major SAC proteins have been identified in *Z. mays*, including MAD2 [103], Bub1, and Bub3 [104], as well as Shugoshin [105] and Aurora B [106].

MAD2 is the central regulator of the spindle assembly checkpoint, and it has been extensively studied in many model systems [107]. MAD2 prevents cells from transitioning from metaphase to anaphase through the indirect inhibition of the anaphase-promoting complex/cyclosome (APC/C) [107]. MAD2 binds Cdc20 along with MAD1 and Bub1, forming the mitotic checkpoint complex (MCC), which prevents Cdc20 from binding and activating the APC/C [108]. This inhibition of APC/C halts the process of cleaving cohesin, preventing chromosome segregation [109]. The maize homolog of MAD2 was identified using alignment analysis and characterized through immunolocalization, revealing a localization pattern on the outer domain of kinetochores (Figure 1A) [103]. This localization pattern increased in the presence of microtubule-depolymerizing drugs, revealing a recruitment to unattached kinetochores [103]. However, localization of MAD2 on meiotic kinetochores was not correlated with the attachment status of the kinetochore, but rather with the tension force. The relative abundance of MAD2 is correlated with the distance between homologous chromosomes (in meiosis I) or sister chromatids (in meiosis II), as great distances correspond to higher tension. The greater the tension, the lower MAD2 staining observed on the kinetochores [103].

Budding inhibited by benzimidazole 1 and 3 (Bub1 and 3) are critical SAC components, where they work in tandem to coordinate checkpoint signaling [110]. In maize, Bub1 is involved in stabilizing kinetochore–microtubule attachments and has been shown to phosphorylate histone H2A (H2A) at threonine 133 (H2AThr133) [104]. The phosphorylated H2AThr133 then associates with CENH3 during meiosis, though this does not appear to impact function during meiosis I [104]. In plants, Bub1 acts on the kinetochores as well, where it acts as a scaffold for coordinating checkpoint signals facilitated by Bub3 [110]. Bub1 is one of the first of the SAC pieces to bind the kinetochore during prophase [111]. From here, Bub1 uses its various domains, N-terminal tetratricopeptide repeat (TPR) domain and blinkin, a member of the conserved KNL1/MIS12 complex/NDC80 complex (KMN) network of kinetochore proteins, to recruit various other SAC proteins [112]. Without Bub3, Bub1 is unable to localize at the kinetochore [113]. Bub1 also interacts with Shugoshin 1 (SGO1) in maize, but no centromere-protective effect is found [104]. Shugoshin plays a crucial role during meiosis by protecting centromere cohesion in maize [104,105,114]. SGO1 functions to prevent separase from releasing sister chromatid cohesion at anaphase I [114]. In mutants where sister chromatids separated during meiosis I, SGO1 is not found [114]. Bub1 plays a larger role in the SAC, where it inhibits the anaphase-promoting complex (APC) when unattached to chromosomes [110].

Aurora B kinase is a highly conserved protein in the spindle assembly checkpoint, and it has been identified in maize based on sequence homology [106]. Broadly, Aurora B plays important roles in kinetochore orientation and connection to microtubules, mainly correcting improper chromosome–microtubule interactions before allowing the cell cycle to proceed [115]. Aurora B is capable of sensing tension on kinetochores to distinguish between properly and improperly attached chromosomes. Correct attachments in which sister chromatids are attached to opposite poles (bi-oriented) generate a tension force that is felt across the kinetochores. Aurora B senses this tension and destabilizes microtubules attachments to the kinetochore that lack tension [116]. Destabilization of these incorrect attachments allows the cell to reattempt proper attachment [117]. In maize, Aurora B kinase has been identified by its 64% sequence similarity to the *Arabidopsis* counterpart [106]. *Z. mays* Aurora B kinase expression is found to be at a similar level to wild-type controls during a low nitrogen condition, and it is hypothesized that Aurora B is not sufficient to correct improper attachments in this context beyond spontaneously occurring attachment errors [106].

## 5. Maize Kinetochore Proteins

Kinetochores are the central hub that integrates all activities involving spindles, chromosomes, and the spindle checkpoint. The kinetochore is a multi-protein structure that assembles on the centromere and facilitates attachment to microtubules in the spindle (Figure 1A) [118]. With this bridging function, individual kinetochore proteins must work together in complex to bind centromeric DNA sequences and assemble a structure capable of attaching to the dynamic ends of microtubules [118]. Spindle assembly checkpoints like Aurora B use the kinetochore as the site of surveillance to ensure proper chromosome attachment, and other checkpoint proteins such as MAD2 use the kinetochore as a site to create the MCC complex that suppresses cell cycle progression [109]. Given the critical role of the kinetochore, the individual proteins that comprise this structure have been extensively studied in various species. In maize, the central proteins that comprise the kinetochore are CENH3, CENPC, NDC80, MIS12, and KNL1 (Figure 1A).

### 5.1. CENH3

First discovered in humans [119], CENH3 (known as CENP-A in metazoans) is a variant of histone protein 3 (H3). Its presence on chromatin defines the centromere independent of specific DNA sequences [120]. Its presence ensures the proper recruitment of other kinetochore proteins, including CENPC, and stabilizes the kinetochore structure [121]. The maize homolog of CENH3 was identified by leveraging the amino acid sequence similarity of CENH3 with the previously identified maize histone H3 protein [122]. Maize CENH3 contains a conserved histone fold domain but has a highly divergent N-terminal tail, which is crucial for its centromere-specific functions [122]. This N-terminal region interacts with the centromeric DNA and contributes to the structural assembly of the kinetochore [122]. A stable CENH3-YFP transgenic line was created and used to identify active centromeres [123]. It was later found that over-expression of CENH3 is lethal to transgenic maize, while over-expression of CENH3 fused proteins such as GFP-CENH3 or CENH3-YFP does not affect their deposition and the growth of transgenic plants [124]. Results from ChIP-seq experiments suggest that CENH3 is essential for recruiting SAC proteins Bub1 and Bub3 to the maize kinetochore [104]. Additionally, centromere failure in maize can result from the dilution of CENH3 during the postmeiotic cell divisions leading up to gamete formation [104]. Another study found a method to bypass cytoplasmic male sterility (CMS) using CENH3 [125]. Cytoplasmic male sterility, a maternally inherited mitochondrial mutation causing sterile pollen, streamlines hybrid maize production by eliminating labor-intensive detasseling for self-pollination control [125]. By generating a frameshift mutation of the CENH3 gene using CRISPR-Cas9, scientists can induce haploid induction, where plants inherit only paternal chromosomes [125]. This allows for a nuclear genome swap from one cytoplasmic background to another [125]. This new method has been extremely successful and has proven useful in the agriculture sector with maize [125]. 

### 5.2. CENPC

CENPC, like CENH3, was first discovered in humans [119]. It anchors centromeric chromatin while recruiting other key kinetochore proteins essential for accurate chromosome segregation [32]. Maize CENPC shares significant similarity with mammalian CENPC and the yeast CENPC homolog, Mif2p [95]. Within its C-terminus, maize CENPC contains a conserved 23-amino-acid sequence known as region I [95]. Mutations in this region lead to significant chromosomal missegregation and mitotic delays [95]. The maize CENPC protein is slightly smaller than human (129kDa vs. 140kDa) [126] but larger than yeast Mif2p (94 kDa) [127]. Maize expresses three CENPC variants: CENPCA, CENPCB, and CENPCC, with CENPCB from a single-copy gene, while CENPCA and CENPCC from multi-copy genes [95]. Although CENPCA and CENPCC genes are 99.9% identical, differences in their C-terminal and 3′ untranslated regions result in proteins that are 95% identical to each other, whereas CENPCB shares 76–78% similarity with both [95]. Immunolocalization experiments reveal that maize CENPC is present in all stages of the cell cycle and localizes to the inner kinetochore domain near the centromere [95]. Electromobility shift assays have demonstrated that maize CENPC directly binds DNA, but this binding interaction is not sequence-specific [121]. Through a series of deletion mapping experiments, a specific 122 amino acid domain coded for in exons 9–12 of the *CENPC* gene was determined to bind both DNA and RNA [121]. The RNA-binding functionality of this domain supports CENPC’s ability to bind centromeric DNA and stabilize the kinetochore [121]. By testing various lengths of RNA, it was shown that longer single-stranded and double-stranded RNAs (44nt) are optimal for CENPC function compared to shorter RNA sequences (24nt) [121]. Using immunostaining and ChIP-Seq, CENH3 and CENPC have been identified on Dp3a, a neocentromere that lacks the typical CentC sequence found in maize centromeres [128]. The findings suggest that a 350kb region on chromosome 3L is sufficient for recruiting kinetochore proteins, offering insights into the initiation of new centromeres during evolution and the role of epigenetics in kinetochore recruitment [128]. These studies highlight the role of CENH3 and CENPC as the main inner kinetochore proteins upon which the rest of the outer kinetochore proteins assemble.

### 5.3. NDC80

NDC80, or Nuclear Division Cycle 80, is one of the four proteins in the NDC80 kinetochore complex that serves as the outer attachment point of the kinetochore with microtubules in the spindle [129]. The NDC80 complex consists of NDC80, Spc25, Spc24, and Nuf2. NDC80 has a molecular weight of 170–190 kDa and is approximately 57 nm long [130]. Ndc80 forms a subcomplex with Nuf2 that is stabilized by a heterodimeric coiled coil [131] and tetramerizes with the Spc24-Spc25 subcomplex [129]. Microtubules connect to kinetochores directly through the NDC80 complex, specifically with the NDC80-Nuf2 subcomplex [132]. The coiled coil domain of the NDC80/Nuf2 heterodimer extends out like a rod that is responsible for direct binding to microtubules [133]. The NDC80 complex functions in tandem with other kinetochore proteins including the Mis12 complex and KNL1 forming the KMN network that stabilize microtubule attachments [133]. NDC80 has a sequence that is highly conserved across many species including yeast, humans, *Xenopus*, and chicken [129]. Sequence conservation facilitated the identification in maize; the NDC80 gene codes for a 576 amino acid protein with a molecular weight of 64kDa [134]. NDC80 localizes outside of CENPC on kinetochores during metaphase II near the microtubule interface [134]. Immunostaining in mitotic root tips revealed that NDC80 assembles on centromeres immediately following replication and is present during all stages of the cell cycle [134].

### 5.4. MIS12

MIS12 (mini chromosome instability 12) is a core kinetochore protein that acts within the KMN network (KNL1, MIS12 and NDC80 complexes) to facilitate interaction with inner kinetochore proteins such as CENPC [135]. MIS12 was one of 12 genes originally identified in a fission yeast mutational screen looking at missegregation of minichromosomes [136]. MIS12 is highly conserved with homologs identified in yeast, humans, *C. elegans*, and chicken [137], and it functions as part of a four-subunit elongated complex [135]. The MIS12 complex is an extended rod composed of two pairs of protein subcomplexes: MIS12:PMF1 and DSN1:NSL [135]. The four-helix bundle attaches to NDC80 at the C-terminal end of the complex [138]. In maize, there are two distinct copies of the *Mis12* gene: *Mis12-1*, with an early stop codon and shorter protein product, and the more highly expressed *Mis12-2* [139]. Immunostaining revealed that both MIS12 variants form a bridge with NDC80 between sister chromatids during meiosis I. This bridge structure allows the two sister kinetochores to act as a single entity during meiosis I when chromatids must stay paired while homologous chromosomes separate [139]. Failure to form the bridge due to RNAi suppression of MIS12 expression resulted in improper chromosome orientation and alignment, which led to premature chromatid and stalling of anaphase I due to unresolved cohesin [139]. Immunostaining revealed that MIS12 proteins are present in the kinetochores of meiotically active cells, localizing specifically to the outer region of the kinetochore during metaphase [139]. Both MIS12 and NDC80 are present throughout the entire cell cycle, whereas the inner kinetochore proteins CENH3 and CENPC are recruited before the cell begins division [139]. Colocalization studies showed that MIS12 sits between inner kinetochore protein CENPC and outer kinetochore protein NDC80, suggesting it serves as a key bridging protein in the maize kinetochore [139].

### 5.5. KNL1

KNL1 is a kinetochore protein that translates the initiation of kinetochore assembly by inner kinetochore proteins into a functional connection for microtubules [133]. KNL1 was initially identified in a *C. elegans* RNAi screen, and its depletion resulted in a kinetochore-null phenotype, showing defects in mitotic chromosome segregation and spindle assembly [140]. Further molecular characterization in *C.elegans* revealed KNL1 to be a 113 kDa protein with weak sequence repeats in the N-terminus and a coiled coil domain in the C-terminus [140]. In the conserved KMN network, KNL1 functions in tandem with NDC80 and MIS12 to build functional kinetochores and form attachments with microtubules [133]. KNL1 interacts directly with MIS12, making a complex that binds cooperatively with the NDC80 complex in vitro [133]. In maize, KNL1 is a constitutive central kinetochore component that recruits protein components of the spindle assembly checkpoint [141]. During division, immunostaining revealed that KNL1 colocalizes with both NDC80 and MIS12 [141] as expected due to the conserved KMN network. While KNL1 is conserved across many species, the maize homolog differs from others, specifically due to the lack of several motifs including SILK, KI, and MELT repeats and the presence of a 630 amino acid disordered domain [141]. Maize KNL1 recruits BMF (Bcl-2-modifying factor) through a hydrophobic domain. This BMF recruitment activity is different from the phosphorylation that occurs in yeast and mammalian cells, which recruit the BMF protein using Mps1-mediated phosphorylation of the MELT repeats that are absent in maize [141]. BMF is a protein in the BH3-only Bcl-2 protein family, which is an essential initiator of apoptosis [142] involved in SAC, specifically monitoring cytoskeletal damage and promoting apoptosis when damage is detected [143].

## 6. Identification of Novel Maize Kinetochore and SAC Proteins

Genomic resources have facilitated the identification of many *Z. mays* genes and their associated proteins based on sequence conservation across species, and molecular tools have facilitated their characterization. Despite the extensive investigations described in this review, many genes whose proteins are involved in chromosome segregation processes remain to be identified and characterized. This is particularly true for genes encoding kinetochore and SAC proteins as the kinetochore structure is known to contain over 100 distinct proteins in humans and over 90 proteins in budding yeast [144]. Even in other plant systems such as *Arabidopsis*, over 20 proteins have been identified, and many of these have not yet been identified in maize including kinetochore proteins CENP-S, CENP-X, KNL2, NNF1, NUF2, SPC24, MPS1, and SAC proteins MAD1, MAD3, INCENP, Borealin, and Survivin [145]. Similarly, kinetochore proteins CENP-S, CENP-X, and CENP-O have been identified in the moss *Physcomitrella patens* but not in maize [145]. To date, the identified and characterized maize proteins include CENH3, CENPC, NDC80, MIS12, KNL1, MAD2, BUB1, and BUB3 as described in this review. Aurora B identification has been suggested [106], but it currently lacks molecular characterization.

Recent efforts to identify the remaining kinetochore and SAC proteins via sequence alignment of both nucleic acid sequence and amino acid sequence has proven challenging. While protein kinetochore function is highly conserved across species, sequence similarity in homologs is too low to allow for identification through basic local alignments. The inability to identify proteins based on basic sequence alignment is a challenge for the broader molecular and genomic fields, and to address this challenge, tools that facilitate structural comparisons are critical. AlphaFold is a machine learning-based computational method that quickly and accurately predicts protein folding, addressing the fifty-year “protein folding problem” [146]. AlphaFold v2.0 was debuted in 2020 at CASP14 (Critical Assessment of Structure Prediction), the international protein folding competition, and it predicted structures with an accuracy near the level of experimentally confirmed structures [147]. Following the competition, DeepMind released the source code to facilitate improvement and usage within different communities [146]. The maize community has directly benefited from this tool as AlphaFold has been integrated into the Maize Genetics and Genomics Database (MaizeGBD) [148] to provide multiple protein structure resources. Two major AlphaFold-based resources include FoldSeek Search Tool for multiple quick views of protein sequence, alignment and structure, and FATCAT Comparison Tool for structural alignments across multiple species [148]. Comparison of protein structure across species has allowed MaizeGDB to make predictions on possible maize homologs of kinetochore and SAC proteins (Table 2).

Predicted protein identities based on structure can be augmented and validated by other approaches including Hidden Markov model (HMM)-based alignments to known proteins. HMMER is a software package (version 3.4, released August 2023) that enables research to model sequence domain families, annotate new sequences, and search for homologous sequences within a defined database [149]. Pairwise sequence alignment programs like BLAST define differences in sequences without accounting for their amino acid positions in relation to the whole sequence, treating each position equally [149]. HMMs, however, use probabilistic methods to represent variation between each amino acid position across a family of homologous sequences. This provides the analysis and identification of even the most distant of homologs, as it captures the evolutionary conservation of a protein domain family [149,150]. HMMs thus provide a broader range of organisms to compare sequences and identify distant or domain-level homologs. We performed HMM-based sequence alignment of AlphaFold predicted kinetochore proteins (as annotated by MaizeGDB) with published kinetochore-specific HMMs created using genome-wide databases, PANTHER11.1, Scope70, pdb70, and Pfam and applied profile-versus-profile searches (Table 2) [151]. HMMER outputs include an E-value, the expected number of false positives or nonhomologous sequences [149]. E-values in HMMER are often debated, but it is generally agreed that values smaller than 10^−3^ indicate true structural homologues and E-values greater than 10^−3^ likely indicate unrelated proteins [149].

HMMER sequence outputs were compared to the MaizeGDB annotations supported by Alphafold predictions (Table 2). In the HMM analysis, we included confirmed maize proteins to serve as controls (indicated with an asterisk in Table 2). In all sequences analyzed, the E-values were significantly below the 10^−3^ threshold, suggesting that the sequences were homologs of known proteins with published HMM models [151]. As illustrated in the table, all HMM-predicted homologs matched the MaizeGBD AlphaFold predicted homologs. This analysis shows that two independent methods, AlphaFold-based structural alignments and HMM-based sequence alignments, achieve the same identification of kinetochore and SAC proteins. In Table 2, there are several novel proteins that have not been previously identified and characterized, specifically SPC25 and NUF2, as well as the Aurora B kinase that lacks specific sequence identification and confirmation. We found two genes (Zm00001eb362200 and Zm00001eb139780) annotated on MaizeGDB as possible SPC25 homologs, and our HMM analysis supports this annotation. Similarly, NUF2 (Zm00001eb064460) has been identified by AlphaFold-based prediction and supported by HMM analysis. Both SPC25 and NUF2 have been identified as members of the NDC80 complex in other species. Additional molecular characterization is needed to confirm these two proteins as localizing with NDC80 in the outer kinetochore of maize. Analyses of Aurora kinase also yielded two potential genes candidates (Zm00001eb121300 and Zm00001eb337510), one of which corresponds with previously identified Aurora B (Zm00001eb121300, identified in [106], while the other is a novel gene sequence. Localization of these proteins to kinetochores and a definitive function in spindle checkpoint function still remains to be determined. The findings displayed in Table 2 show that new tools will facilitate the sequence and structure-based identification of protein in *Z. mays*, with many proteins involved in chromosome segregation yet to be identified and characterized.

**Table 2 genes-15-01606-t002:** Maize sequence predictions based on structure and sequence. The *Z. mays* gene sequence IDs used in HMMER analysis are displayed in the second column. The third column displays the MaizeGDB AlphaFold predicted protein identity. The fourth column displays the HMMER-predicted protein identity based on kinetochore-specific models from Tromer et al. [151] and the associated E-values for this prediction in the last column. The asterisk indicates genes that have been annotated with protein identity in MaizeGDB.

Type of Protein	Maize Sequence	MaizeGDB	HMMER Protein	E-Value
Spindle Assembly Checkpoint	Zm00001eb425240 *	MAD2	MAD2	3.30 × 10^−83^
	Zm00001eb121300 *	Aurora B kinase 1	Aurora kinase	1.40 × 10^−142^
	Zm00001eb337510	Aur1 like	Aurora kinase	5.40 × 10^−142^
Kinetochore	Zm00001eb291210 *	CENH3	CENPA	3.00 × 10^−17^
	Zm00001eb433940 *	MIS12-1	MIS12	7.30 × 10^−29^
	Zm00001eb065850 *	MIS12-2	MIS12	4.50 × 10^−32^
	Zm00001eb064460	NUF2	NUF2	1.30 × 10^−88^
	Zm00001eb362200	SPC25	SPC25	1.60 × 10^−46^
	Zm00001eb139780	SPC25	SPC25	6.90 × 10^−50^

## 7. Conclusions

*Z. mays* has a long history as a genetic model organism, particularly in cytogenetic studies with its large chromosomes and microscopy-based tools. Modern advances in fluorescence microscopy- and -omics-based tools including a high-quality genome and protein predictions has furthered investigation in maize, particularly in the area of chromosome segregation. Studies in *Z. mays* have expanded our understanding of the dynamics, regulation, and positioning of acentrosomal spindles. With its Abnormal 10 variant, maize provides an unparalleled system to investigate and understand meiotic drive systems that break the inheritance predictions of Mendelian genetics. The spindle assembly checkpoint that monitors chromosomal attachments to the spindle and regulates the attachment of microtubules to the kinetochore has been studied in maize. The main SAC effector proteins MAD2, Bub1, and Bub3 have been identified, but many SAC proteins remain unidentified in maize. Similarly, five major kinetochore proteins (CENH3, CENPC, MIS12, KNL1, NDC80) have been identified in maize, but with over 100 kinetochore proteins identified in animal systems, many maize kinetochore proteins remain to be identified and characterized. New protein identification tools that utilize three-dimensional structure and domain conservation will facilitate identification and characterization of these proteins critical for chromosome segregation in *Z. mays*.

## Figures and Tables

**Figure 1 genes-15-01606-f001:**
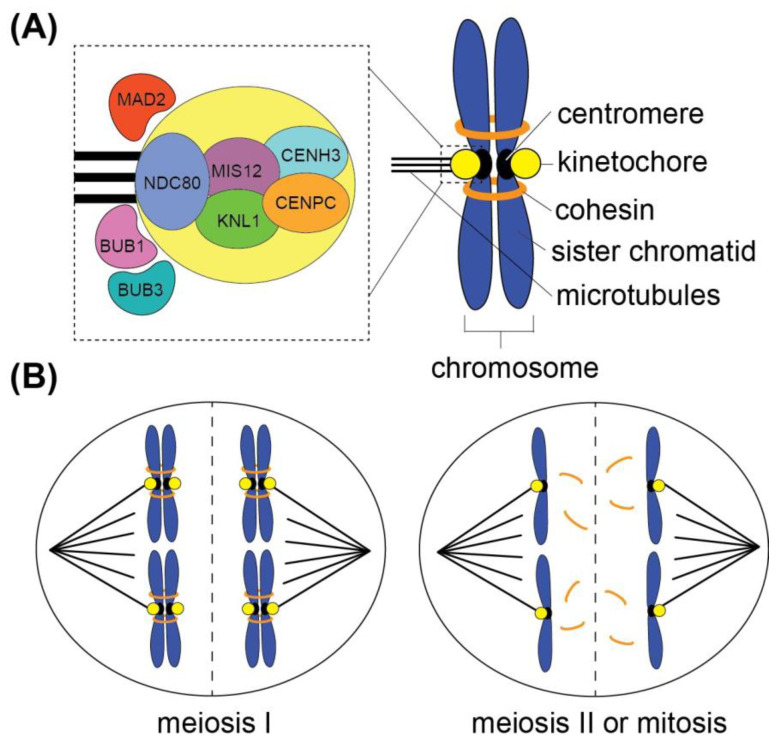
Chromosome segregation machinery. The segregation machinery that separates chromosomes in mitosis and meiosis is the spindle, a microtubule-based structure. (**A**) Spindle microtubules attach to kinetochores, multi-protein structures that assemble on centromeres. The dashed box shows a zoomed in image of a kinetochore and includes the proteins identified in *Z. mays* (kinetochore proteins NDC80, MIS12, KNL1, CENPC and CENH3, and spindle checkpoint proteins MAD2, BUB1, and BUB3 that localize on the outer kinetochore). Replicated chromosomes contain two sister chromatids held together by cohesin until cleavage in mitotic anaphase or meiotic anaphase II. (**B**) In meiosis I, the spindle segregates homologous chromosomes, and in meiosis II and mitosis, the spindle segregates sister chromatids after cohesin is degraded. The dashed line indicates the plane of cell division.

**Figure 2 genes-15-01606-f002:**
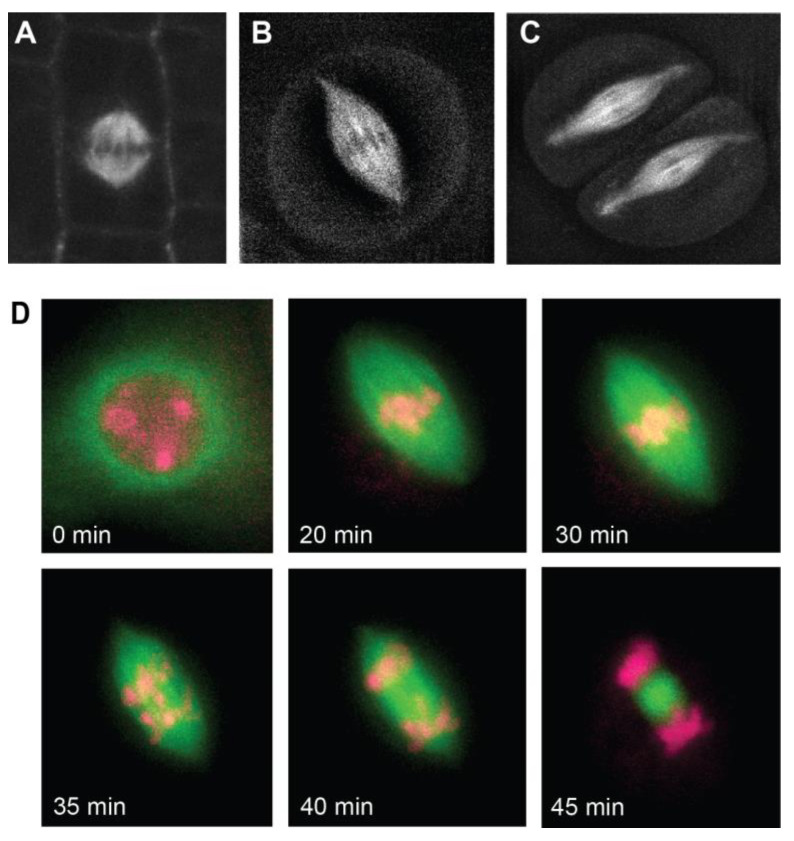
*Zea mays* spindles. Live images of *Z. mays* spindles can be acquired due to the development of fluorescently tagged tubulin. In all images, B-tubulin was tagged with a florescent protein to facilitate imaging. (**A**) Maize mitotic spindle; microtubules are shown in white (image courtesy of Carolyn G. Rasmussen). (**B**) Maize meiosis I spindle; microtubules are shown in white. (**C**) Maize meiosis II spindles, microtubules are shown in white. There are two cells, each containing a meiotic spindle. (**D**) Time course of meiosis I spindle assembly (0–20 min), alignment of chromosomes on the metaphase I spindle (30 min), and anaphase I segregation of chromosomes (35–45 min). Microtubules are shown in green and chromosomes (labelled with SYTO12 DNA stain) are shown in pink. At time point 0 min, the nuclear envelope is still intact and microtubules can be seen encircling the nuclear membrane. At time point 45 min, the spindle has disassembled and the remaining microtubule structure between the separated chromosomes is the phragmoplast.

**Figure 3 genes-15-01606-f003:**
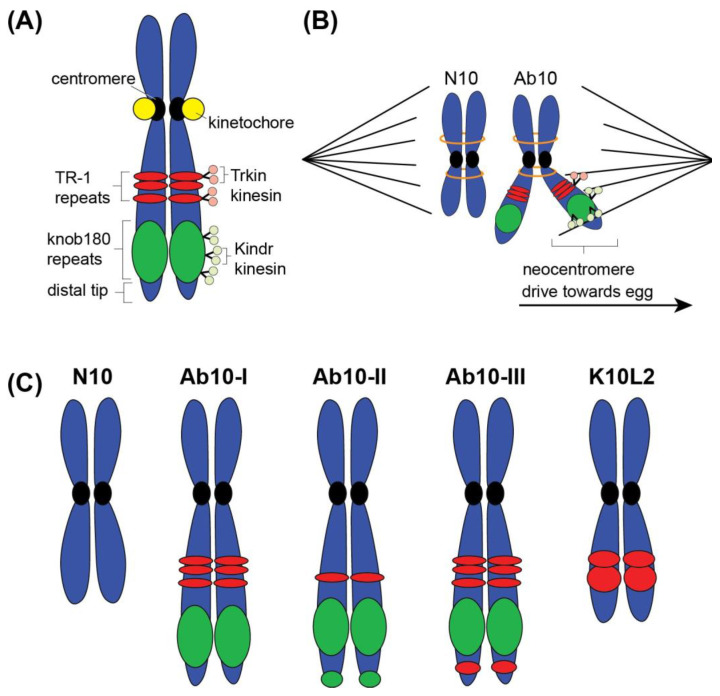
Abnormal chromosome 10 (Ab10) structure and function in maize meiotic drive. (**A**) Ab10 differs from N10 due to the presence of heterochromatic repetitive DNA sequences, TR-1 (shown in red), and knob180 (shown in green). Ab10 also contains genes that code for the TRKIN and KINDR kinesins that associate with TR-1 and knob180 sequences, respectively. (**B**) The kinesins interact with the TR-1 and knob180 sequence to create neocentromere function capable of binding microtubules and pulling Ab10 chromosomes preferentially toward the developing egg cell. This neocentromere activity is functional before kinetochores are operational on centromeres, thus giving Ab10 chromosomes an advantage due to early interaction with spindle microtubules. (**C**) There are three distinct Ab10 types (type I, II, III) that can be cytologically distinguished from N10 due to the presence of TR-1 and knob180 sequences. K10L2 is another chromosome 10 variant that is unique from both N10 and Ab10. All variants show differing rates of meiotic drive, likely due to the difference the amount and location of knob DNA sequences.

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
