# Peer review of "Mechanisms, Machinery, and Dynamics of Chromosome Segregation in Zea mays"

_genes, 2024, doi:10.3390/genes15121606_

Round 1
Reviewer 1 Report
Comments and Suggestions for Authors
The work is aimed to provide a review about the advances made in maize chromosome segregation.
The work is well organized and the contents are discussed in detail.
The conclusions are consistent with the aims of the work.
However, the following revisions are suggested:
• In the abstract, the value added of the work compared to previous reviews about the discussed topic could be included
• In the “Introduction” section,
o Please reformulate the sentence at Lines 38-39
• In the 3.1 section
o The recent review work available at https://par.nsf.gov/servlets/purl/10439783In could be mentioned
A careful rereading of the manuscript is suggested, in order to avoid repetition of terms and make some sentences clearer.
Some minor issues:
· Please check the format of the manuscript, including the references format, https://www.mdpi.com/journal/genes/instructions
Author Response
Response to Reviewer 1
We thank the reviewer for their careful reading of our manuscript and helpful feedback. We are addressed the reviewers suggested in the following ways:
Reviewer suggestion: “In the abstract, the value added of the work compared to previous reviews about the discussed topic could be included”
We have added some new language in the Abstract, specifically calling out that our review is unique compared to others in the field given that we take a species-specific approach discussing Z.mays features of chromosome segregation including points that are unique to maize, but also features that are conserved with other species. We also clarified our final sentence, highlighting our contribution to the field by identifying novel maize kinetochore proteins.
Reviewer asked: “In the “Introduction” section, please reformulate the sentence at Lines 38-39”
We have re-written this sentence clarify the number of genes and their organization. The sentence now reads: “The current Z. mays genome assembly (Zm-B73-REFERENCE-NAM-5.0) is composed of 2.3 Gb base pairs with approximately 33,000 protein-coding genes organized into ten chromosomes [7].”
Reviewer asked: “ In the 3.1 section, the recent review work available at https://par.nsf.gov/servlets/purl/10439783 could be mentioned”
We have added a sentence at the end of the opening paragraph in Section 3 to acknowledge and cite Kelly Dawe’s Ab10 review article: “A recent review article by R. Kelly Dawe delves deeply into the history and mechanism of maize meiotic drive [81]; in this review, we will summarize the major features of this drive system. “
Reviewer suggestion: A careful rereading of the manuscript is suggested, in order to avoid repetition of terms and make some sentences clearer. Some minor issues: Please check the format of the manuscript, including the references format, https://www.mdpi.com/journal/genes/instructions
We have carefully reviewed the manuscript and clarified writing, removed repetitive sequences and ensured that our reference format matches the instructions.
Reviewer 2 Report
Comments and Suggestions for Authors
The paper is interesting, and its information is sound and valuable for the community. The main problem is that it is only a text review with a single table. This fact needs to improve the impact and usefulness. So I recommend:
Include a table summarizing the main findings and the key citations. That is
Dynamic Assembly of the Spindle; Abnormal Chromosome 10 (Ab10); Kinetochore Function; Spindle Assembly Checkpoint; Molecular Tools and Genomic Resources:
Include a graphic explaining how Zea mays' spindle assembly checkpoint and kinetochore proteins function.
Include a table summarizing the new software tools helpful in studying chromosome segregation in maize and their advantages and disadvantages.
With all these additions, the review will be more useful.
Author Response
Response to Reviewer 2
We thank the reviewer for their careful reading of our manuscript and helpful feedback. We are addressed the reviewers suggested in the following ways:
Reviewer suggestion: Include a table summarizing the main findings and the key citations. That is Dynamic Assembly of the Spindle; Abnormal Chromosome 10 (Ab10); Kinetochore Function; Spindle Assembly Checkpoint; Molecular Tools and Genomic Resources.
We agree with the reviewer that additional tables and figures would improve the manuscript. We have created a table (Table 1) with the five major topic areas suggested by the reviewer and include the major highlights in each area. We also direct readers within the table to new figures created to help clarify the topics discussed.
Reviewer suggestion: Include a graphic explaining how Zea mays' spindle assembly checkpoint and kinetochore proteins function. Include a table summarizing the new software tools helpful in studying chromosome segregation in maize and their advantages and disadvantages
In addition to Table 1 (includes software, tools and resources), we have created three figures:
Figure 1: a diagram of maize kinetochores and spindle, and a diagram of meiosis vs. mitosis chromosome segregation
Figure 2: live microscopy images of maize mitosis (courtesy of Carolyn Rasmussen), meiosis I and meiosis II spindles, and live imaging timepoints of maize meiotic spindle assembly and chromosome segregation
Figure 3: a diagram of Abnormal Chromosome 10 structure, neocentromere function, and Ab10 types
Reviewer 3 Report
Comments and Suggestions for Authors
In this review, the authors have explored advancements in our understanding of how Zea mays chromosomes are segregated by spindle and kinetochore machinery. I recommend the authors provide more detailed explanations of the mechanisms underlying these processes. Additionally, I suggest including relevant figures and tables to better illustrate the key concepts, particularly focusing on chromosome dynamics and segregation. These additions would enhance the clarity and comprehensiveness of the review.
Author Response
Response to Reviewer 3
We thank the reviewer for their careful reading of our manuscript and helpful feedback. We are addressed the reviewers suggested in the following ways:
Reviewer suggestion: "I recommend the authors provide more detailed explanations of the mechanisms underlying these processes."
We have edited the manuscript to include additional clarifying descriptions of mechanisms, but we also would like to keep the review focused on the specific information and studies that have been performed in Z. mays, which less well-studied than many other model organisms such as yeast, mouse, Drosophila and Arabidopsis in terms of molecular mechanisms underlying these processes. We believe this is one of the strengths of the review; it is often challenging to determine from reading the literature what advances are specific to maize vs. plants in general. For example, the recent review by Bo Liu and Yuh-Ru Lee (“Spindle Assembly and Mitosis in Plants”, 2022) is an excellent review of segregation processes and mechanisms, but difficult to separate maize-specific discoveries vs. other plants.
Reviewer suggestion: "Additionally, I suggest including relevant figures and tables to better illustrate the key concepts, particularly focusing on chromosome dynamics and segregation."
We agree with the reviewer that additional figures are needed so we have added one additional table (Table 1) that summarizes the five major topic areas of the review and includes the major highlights in each area. We also created three figures:
Figure 1: a diagram of maize kinetochores and spindle, and a diagram of meiosis vs. mitosis chromosome segregation
Figure 2: live microscopy images of maize mitosis (courtesy of Carolyn Rasmussen), meiosis I and meiosis II spindles, and live imaging timepoints of maize meiotic spindle assembly and chromosome segregation
Figure 3: a diagram of Abnormal Chromosome 10 structure, neocentromere function, and Ab10 types
Round 2
Reviewer 2 Report
Comments and Suggestions for Authors
The table and figures have significantly enhanced the review. I can endorse the publication